# Exploring Macro Discourse Structure with Macro-micro Unified Primary-secondary Relationship

## Abstract

Discourse structure analysis is helpful for the machine to identify different types of discourse writing styles, and lays the foundation for the study of discourse automatic generation. In this paper, after studying the difference and the relationship between micro and macro discourse structures, we explore the macro discourse structure, and put forward a Macro Chinese Discourse Treebank (MCDTB) on the top of existing Chinese Discourse Treebank (CDTB), by unifying micro and macro discourse structures. Specifically, at the micro level, we put forward the primary-secondary relationship from the logical semantic perspective, while at the macro level, we put forward the primary-secondary relationship from the pragmatic function perspective. Preliminary experimentation shows that our macro-micro unified schema is appropriate for the discourse structure analysis.

## 1 Introduction

Discourse rarely exists isolated, so the interpretation of a discourse requires understanding of its structure and semantics. (De Beaugrande, 1981) As an instance, the discourse primary-secondary relationship is the relation between the primary content and secondary content within a discourse, or the relation between the primary and secondary aspects of a discourse and the others. Here, the primary content refers to the dominant position in the discourse, and plays a decisive role in the part, while the secondary content refers to the auxiliary position, and does not play a decisive role in the discourse.

In principle, the primary-secondary relationship plays a critical role in natural language processing, since the recognition of the discourse primary-secondary relationship not only helps to understand the discourse structure and semantics, but also provides strong support for deep natural language processing applications. Despite of its potential in statistical machine translation (Meyer and Popescu-Belis, 2012; Guzmán et al., 2014; Peldszus and Stede, 2015), automatic text summarization (Atkinson and Munoz, 2013; Ferreira et al., 2014; Cohan and Goharian, 2015), question answering (Liakata et al., 2013), information extraction (Presutti et al., 2012; Zou et al., 2014), sentiment analysis (Mukherjee et al., 2012; Mittal et al., 2013; Bhatia et al., 2015) etc. recognition of the discourse primary-secondary relationship has become the bottleneck in discourse structure analysis recently, largely due to the ignorance of its critical role in discourse structure analysis by viewing it only as a dispensable component in the analysis of the discourse rhetorical structure.

Generally speaking, there exist two hierarchical levels of discourse structures: micro level and macro level. While micro structure refers to the relationship between the internal structures in a sentence or two consecutive sentences, the macro structure refers to the relationship between sentences, paragraphs and chapters, at a higher level.

At the micro level, the Rhetorical Structure Theory (RST) (Mann and Thompson, 1987), one of the most influential discourse theories, represents the discourse structure as a tree with phrases or clauses as elementary units(EDUs), where adjacent EDUs are connected by rhetorical relations, forming larger discourse units. Although RST establishes two different types of units, where nucleus are considered as the central parts, and satellites as peripheral ones. The nuclearity defined in RST is considered from the particular rhetorical relation, instead of the overall discourse, and with-

out considering the overall discourse intention, the nuclearity is difficult to distinguish.

At the macro level, Van Dijk (1980) proposes the macrostructure theory, indicating that a text not only has local or microstructure relationship between subsequent sentences, but also has overall structure that defines their global coherence and organization. Although Van Dijk explores the macrostructure from both semantics and pragmatics, he focuses on the semantic aspect of the macrostructure, emphasizing that without the semantic macrostructure, it is unable to explain the meaning of the overall discourse and understand the partial coherence in the macrostructure. Unfortunately, no matter whether at micro level or at macro level, existing theories focus on logical semantics, with few from the pragmatic function perspective.

In this paper, we regard the recognition of primary-secondary relationship as an independent task from the discourse rhetorical structure analysis, and integrate the overall discourse with the primary-secondary relationship as the unified carrier. For this purpose, a macro discourse structure representation schema is proposed. At the micro level, we lay on the logical semantic relationship between the discourse units based on the rhetorical structure, while at the macro level, we lay on the pragmatic function of the content based on the schematic structure. Specifically, our work expands the discourse analysis from intra-paragraph to the overall discourse. Guided by the representation schema, we annotate 97 news wire articles both from the perspective of micro semantics and macro pragmatics. Table 1 compares the macro and micro structures. Preliminary experiments on the recognition of discourse relation and primary-secondary relation show that our macro-micro unified representation schema is appropriate for the discourse structure analysis tasks.

## 2 Related work

In this section, we first summarize theories wildly employed in discourse structure, then we present existing approaches to discourse structure analysis.

### 2.1 Discourse Theory

From micro perspective, discourse structure theories mainly include Cohesion Theory(Halliday, 1971), Hobbs Model(Hobbs, 1993, 1979), Rhetor-

ical Structure Theory(Mann and Thompson, 1986, 1987; Mann et al., 1992), Intention Structure Theory (Grosz and Sidner, 1986; Grosz et al., 1995), Chinese Complex Sentences Theory (Xing, 2001; Yao, 2006), Connective-driven Dependency Tree Theory (Li, 2015; Li et al., 2014) etc. From macro perspective, the theories of discourse structure are relatively few, including Super-theme Theory (Martin and Rose, 2003), Macrostructures Theory (Van Dijk, 1980, 1985, 2013) etc.

**Cohesion Theory:** Halliday (1971) pointed out in cohesion theory, "cohesion considers the interaction of cohesion with other aspects of text organization." When Grimes (1975) depended the cohesion theory, he identified two types of functional relations: 1) paratactic, where all parts of a relation are equally prominent in terms of their discoursal functions; and 2) hypotactic, where one part of a relation is more prominent than the other part.

**Rhetorical Structure Theory:** RST (Mann and Thompson, 1986) provides a general way to describe the relations among the elementary discourse units, and establishes two different types of units, nucleus and satellite. Nuclei are considered as the most important parts of text whereas satellites contribute to the nuclei and are secondary. Figure 1 shows the rhetorical structure of three types of rhetorical relation: (a) evidence; (b) circumstance; and (c) sequence. The arrows point to the nuclei in the structure.

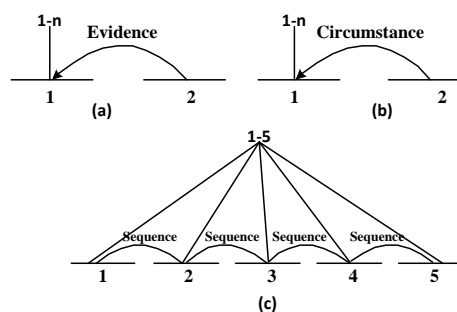

Figure 1: Rhetorical Structure Theory

**Connective-driven Dependency Tree:** Li et al. (2014) proposed a Connective-driven Dependency Tree(CDT) scheme to represent discourse rhetorical structure, with EDUs as leaf nodes and connectives as non-leaf nodes, to accommodate the special characteristics of Chinese language. A discourse can be expressed to a complete discourse structure tree based on the CDT representation schema.

| Perspective | Structure | Concern | Description |
|---|---|---|---|
| Micro | Rhetorical Structure | Logical semantics | Structural analysis from the form |
| Macro | Schematic Structure | Functional Pragmatics | Functional analysis from the information |

Table 1: Comparison of micro and macro structures

**Macrostructure Theory:** The point of macrostructures is that texts not only have local or micro structural relations between subsequent sentences, but also have overall structures that define their global coherence and organization (Van Dijk, 1980). Figure 2 shows Van Dijk's macrostructure theory. In this marcostructure, $P_i$ represents a lower layer proposition, and $M_j^k$ represents a marcostructure unit, in which the superscript $k$ represents the hierarchy, while the subscript $j$ indicates the unit order on current layer.

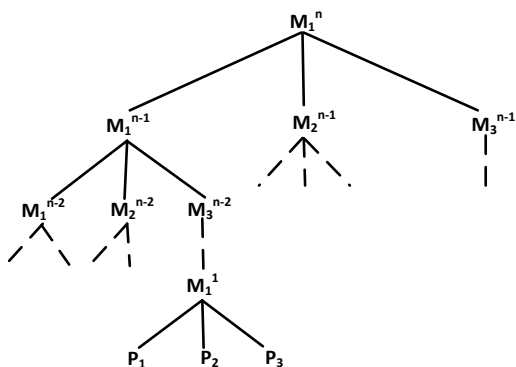

Figure 2: Macrostructure Theory

### 2.2 Computational Models

Since Marcu (2000) first developed a rule-based discourse parser, several algorithms for discourse parsing have been proposed, both statistical and rule-based, especially since the release of English discourse corpora such as Penn Discourse Treebank (PDTB) and the Rhetorical Structure Theory Discourse Treebank (RST-DT)

For the research in RST-DT, SPADE(Soricut and Marcu, 2003) is a sentence level discourse parser. Two probabilistic models were employed that use syntactic and lexical information to segment and parse text. LeThanh et al. (2004) proposed a multi-step algorithm to segment text and organize the spans into trees for each successive level of text organization. Hernault et al. (2010) presented HILDA, an implemented discourse parser by combining adjacent spans with

greedy bottom-up tree building approach, they built discourse trees in linear time complexity with respect to the length of the input text. Joty et al. (2013) built a discourse tree by applying an optimal parsing algorithm to probabilities inferred from two Conditional Random Fields: one for intra-sentential parsing and the other for multi-sentential parsing. The performance of the algorithms we mentioned above is shown in Table 2.

On the contrast of the researches at the micro level, very few corpus resources and computational models are proposed at the macro level.

| Reference | Primary-secondary relationship | | | Discourse structure |
|---|---|---|---|---|
| | Precision | Recall | F-score | F-score |
| S & M | 54.0% | 21.6% | 30.9% | 49.0% |
| Le Thanh | 47.8% | 46.4% | 47.1% | 53.7% |
| Hernault | 61.4% | 61.2% | 61.3% | 47.3% |
| Joty | - | - | 68.4% | 55.7% |

Table 2: Primary-secondary relationship recognition in English discourse corpus RST-DT
S & M: (Soricut and Marcu, 2003); Le Thanh: (LeThanh et al., 2004); Hernault: (Hernault et al., 2010); Joty: (Joty et al., 2013)

## 3 Macro Discourse Representation Schema

The overall discourse structure is relevant to the discourse genre and discourse pattern, thus discourse structures vary if the genres are different. For example, the news articles are commonly described in *summary-story* structure, and academic papers are consist of *abstract, introduction, related work, experimentation, conclusion* etc. while the court documents are recorded in the structure of *in what way, for what reason, where, according to what inference* etc.

There are also different ways of discourse expression for the same genre, that is, different discourse patterns. The discourse pattern is the macro structure of text organization, and related to the specific environment and use habits, after repeated use. Various genres formed their own particular and stylized discourse organization structure and characteristics, and became

the rules obeyed by the same domain. According to linguists, the common discourse patterns include *Problem-Response Pattern*, *Question-Answer Pattern*, *Claim-Counterclaim Pattern*, *General-Particular Pattern* etc. (Hoey, 2001)

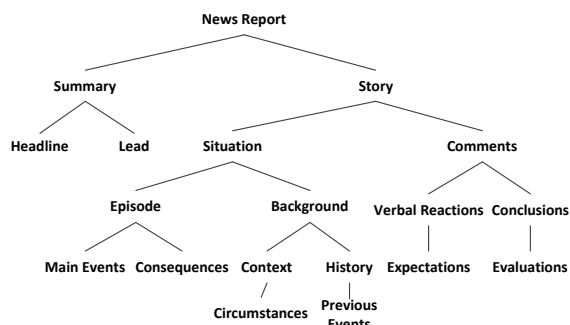

Figure 3: Schematic Structure on News

Figure 3 is the schematic structure of news genre(Van Dijk, 2013). There are two main parts in the structure: summary and story. Summary contains headline and lead, while story includes situation and comments, and all these parts together constitute the macrostructure of news discourse through a top-down hierarchical structure. Such a schematic structure allows readers to quickly analyze and understand the discourse function in the communicative situation.

Inspired by Van Dijk's schematic structure, we unify the primary-secondary relationship which defined in RST (nuclearity) and macrostructure separately, and combine the micro and macro discourse structure together, and then explore a Macro Discourse structure representation schema. In this representation schema, discourse structure tree is represented the hierarchical discourse(as shown in Figure 4). We construct a top-down structure with multiple layers, composed of title, chapter, paragraph, microstructure, and elementary discourse unit. In general, there is only one layer of the title and the chapter, while the other structural layers are composed of multiple layers. The arrow direction indicates the primary unit of discourse relation. For the microstructure, we follow the definition of CDT, which not expanded in the Figure 4.

We expand the discourse analysis from intra-paragraph to the overall discourse on the basis of original discourse structure analysis. At the micro level, we analyze the logical semantic relationship between the discourse units based on the rhetorical structure; at the macro level, we analyze the function of the discourse content based on the schematic structure. Guided by this representation, we take the genre of news wire as the main research object, and annotate 97 news wire articles both from the perspective of micro semantics and macro pragmatics.

For example, in the case of magazine article chtb_0596 from Chinese Treebank 8.0, the macro schematic structure and the micro rhetorical structure of our annotation are shown in figure 5 and figure 6. In schematic structure, we annotate the function of each paragraph and unit group, while in rhetorical structure we focus on the rhetorical relation between discourse units.

### 3.1 Leaf Nodes

Unlike the definition in microstructure (the elementary units are treated as leaf nodes), on the macro layer, we directly treat the paragraphs which are natural segmented in the discourses as leaf nodes.

Take the chtb_0019 for example, which is a typical news wire article from CTB 8.0. There are five paragraphs in the news *Significant achievements in the construction of Ningbo Free Trade Zone*, so we treat the five paragraphs(1,2,3,4,5) as leaf nodes directly, and the rhetorical structure of this discourse is shown as Figure 7. Limited to the length of this paper, the full discourse text of this example is not included, please refer to the corpus CTB 8.0.

### 3.2 Non-leaf Nodes

Discourse relations connect discourse units, which are treated as non-leaf nodes in our macro discourse structure. In our representation scheme, we divide the discourse relations into three categories, fifteen subcategories which listed in Table 3. As shown in Figure 7, the relations *Elaboration*, *Background*, and *Joint* are non-leaf nodes in the structure tree.

### 3.3 Primary-secondary Relationship

A discourse relation generally includes two or more discourse units, these discourse units belong to the same relation layer. If one of the discourse units can generalize the intention and content of the relation layer, and can represent the relation layer connect to other layers, this discourse unit is a primary unit, while others are secondary ones. If the two or more discourse units are equally important, this relation is a multi-primary relation.

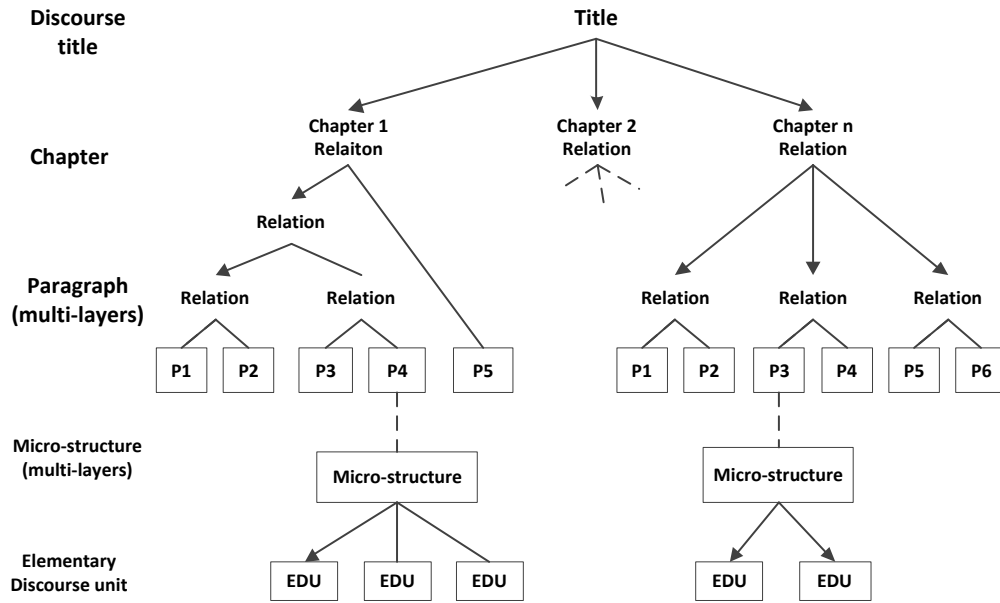

Figure 4: Macro Discourse Structure

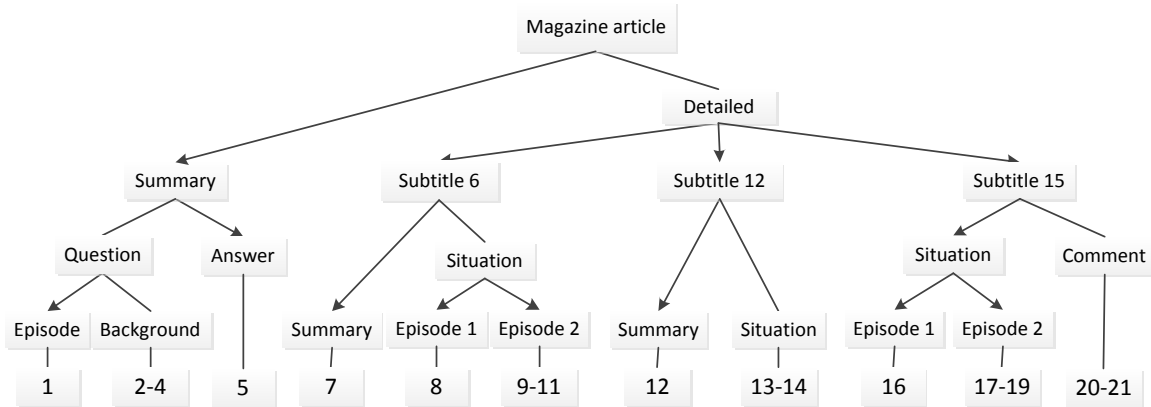

Figure 5: Macro Schematic Structure of chtb_0596

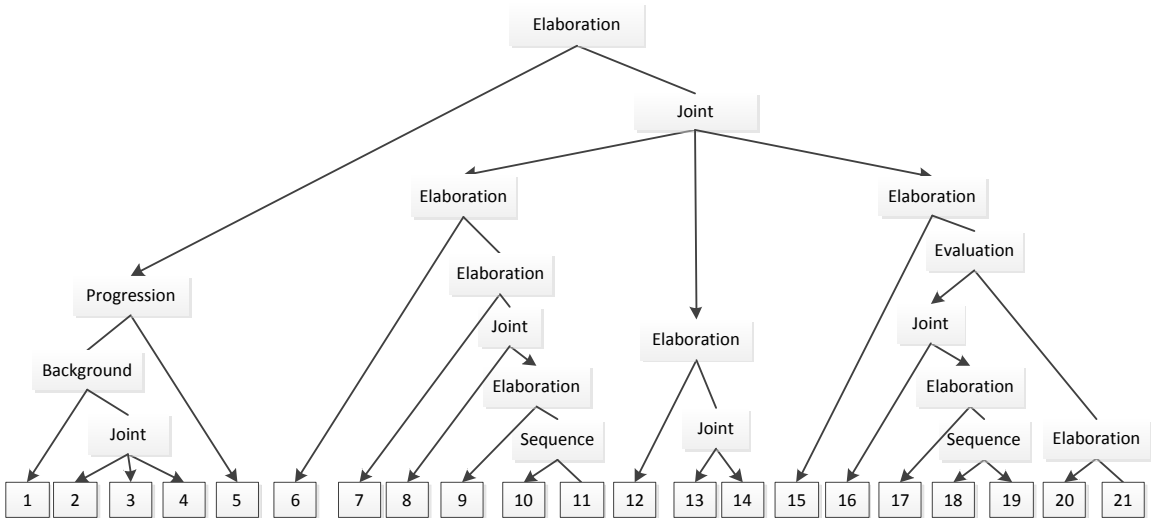

Figure 6: Micro Rhetorical Structure of chtb_0596

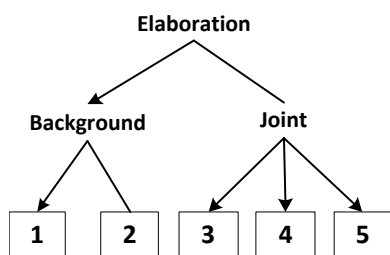

Figure 7: Micro Rhetorical structure of chtb_0019

| Categories | Subcategories |
|---|---|
| Coordination | Joint |
| | Sequence |
| | Progression |
| | Contrast |
| | Supplement |
| Causality | Cause-Result |
| | Result-Cause |
| | Behavior-Purpose |
| | Purpose-Behavior |
| | Background |
| Elaboration | Elaboration |
| | Statement-Illustration |
| | Illustration-Statement |
| | Summary |
| | Evaluation |

Table 3: Micro Relationship in our corpus

We define three types of primary-secondary relations:(1) P-S, the former unit is primary, and the latter unit is secondary; (2) S-P, the former unit is secondary, and the latter unit is primary; (3) MP, multi-primary. For instance, in statement-illustration relation, one of the discourse units is statement, while the other is illustration, the illustration unit is a service for the statement, so the statement-illustration relation is P-S relation; another example, in joint relation, there can be two or more discourse units, one or more discourse units may act as primary parts, so joint relation may be a P-S or S-P or MP relation.

From the micro perspective, the primary-secondary relationship represents the relation between sentences or sentence groups, while from the macro perspective, the primary-secondary relationship represents the relation between paragraphs, chapters or discourses.

In the discourse structure tree shown in Figure 7, the arrows point to the primary units in the discourse. In this discourse, paragraph 2 introduces the approval and development situation of *Ningbo free trade zone*, which is the background of the event *Ningbo free trade zone has achieved fruitful results* mentioned in paragraph

1. These two paragraphs constitute a background relation, in which paragraph 1 is a primary unit, and paragraph 2 is a secondary unit according to the content. Paragraphs 3, 4 and 5 respectively from the three aspects expounds the *fruitful results* mentioned in paragraph 1, the discourse relation among them is joint, and all the three paragraphs constitute a whole unit, explain the overall structure constituted by paragraph 1 and 2, the two paragraph groups form the relation of elaboration, in which the paragraph 1 and 2 is more important. In the overall discourse, paragraph 1 can best express the discourse topic *Significant achievements in the construction of Ningbo Free Trade Zone* (also the discourse title), paragraph 1 is therefore the most important paragraph among all these discourse units. Based on the arrow pointing in the discourse structure tree can also get the same conclusion.

### 3.4 Macro Schematic Structure

In the representation schema, the discourse is organized as a tree structure, in which paragraphs appear in the leaf nodes and the discourse relations appear in the non-leaf nodes. The tree structure is an appropriate representation of discourse structure, which not only expresses the hierarchical relationship of the discourse, but also expresses the primary-secondary relationship between the discourse units. Essentially, the depth of the hierarchical structure indicates the depth of the corresponding discourse semantic.

Based on the micro rhetorical discourse structure, we convert the logical semantic relationship to pragmatic function relationship, and establish the schematic structure. We analyze and understand the discourse from a global perspective, and add the pragmatic function to both the leaf nodes and non-leaf nodes, which means all the discourse units on each layer. The functions of every discourse unit form a completed schematic structure tree. Figure 5 and Figure 6 respectively show the macro schematic structure and micro rhetorical structure.

## 4 Building a Macro Chinese Discourse Treebank

It is necessary to construct a micro and macro unified discourse corpus for discourse structure analysis. Annotation research of macro and micro unified discourse structure, will not only lay the foun-

dation for the research of discourse structure analysis, but also provide strong support for deeper NLP applications.

### 4.1 Annotation Overview

Guided by the Macro Discourse Structure, we annotate a Macro Chinese Discourse Treebank(MCDTB) consist of 97 Xinhua news wire articles. Preliminary experiments show that the presentation schema and corpus resource are appropriate to the macro discourse structure analysis.

Because the discourse units are not isolated from the overall discourse, its difficult to judge whether the discourse units are important or not simply from the units themselves. It is necessary to have a comprehensive understanding of the overall article when judging the primary-secondary relations and functional roles.

In determining primary-secondary relations, we should have a full understanding of the discourse intention and main content, and grasp the principle of local and global. Local principle refers to the primary units could summarize the main content and intention of its relation layer, and represent the relation layer relating to the context; global principle refers to the primary discourse units should conform to the main intention of the overall article.

We annotate the discourse topic, lead, abstract, paragraph topics, discourse relations, primary-secondary relations, paragraph segments, and explore the relationship between microstructure and macrostructure.

### 4.2 Annotation Strategy

We employ a combination of top-down and bottom-up strategy in the annotation work. (1) We determine the overall level first and then analysis goes on step by step to the individual discourse units. Such a top-down strategy can easily grasp the overall discourse structure, which consistent with the reading habit of human beings. (2) Meanwhile, we determine whether the lower discourse units need to be combined first according to the similarity of their forms and contents, and combine them together as a whole unit to contact with other parts. Our annotation work shows the strategy is effective.

### 4.3 Quality Assurance

To ensure the quality of our corpus, we adopt the annotator consistency using agreement and kappa

on 20 documents(chtb_0011-chtb_0030). Table 4 illustrates the annotation consistency in detail.

|  | Agreement | Kappa |
| --- | --- | --- |
| Discourse structure | 84.2% | 0.68 |
| Discourse relation | 79.0% | 0.69 |
| Primary-secondary relation | 78.9% | 0.68 |

Table 4: Annotation Consistency

As shown in Table 4 we measure the agreement and Kappa of discourse structure, discourse relation, and primary-secondary relation. Its very difficult to achieve high consistence because the judgments of relation and structure are very subjective. Our measurement data is only taken on the layer of leaf nodes.

### 4.4 Corpus Statistics

Our corpus consists of 97 news wire articles from Chinese Treebank 8.0. There are 533 paragraphs with 438 discourse relations, and each article with an average of 5.49 paragraphs. Detailed statistical data are shown in Table 5.

| Count of documents | 97 |
| --- | --- |
| Count of paragraphs | 533 |
| Average paragraphs in document | 5.49 |
| Maximal of paragraphs | 13 |
| Minimal of paragraphs | 2 |
| Count of sentences | 1339 |
| Average paragraph length (sentences/ paragraph) | 2.51 |
| Amount of macro discourse relations | 438 |

Table 5: Corpus Statistics

We also count the primary-secondary data for each subcategory (shown as Table 6). Among the 438 relations, 151 are joint and all of them are multi-nucleus; 121 are elaboration and all of them have a nucleus in the former discourse unit. The statistics shows the relationship between discourse relation and primary-secondary relation is strong dependence, we can make further exploration on this phenomenon later.

## 5 Preliminary Experimentation

Based on the corpus resource we built, we can do the following analysis: discourse relation discovery, discourse relation recognition, primary-secondary relation recognition, and discourse

| Discourse relation | Total | P-S | S-P | MP |
|---|---|---|---|---|
| Joint | 151 | 0 | 0 | 151 |
| Sequence | 17 | 1 | 1 | 15 |
| Progression | 2 | 0 | 1 | 1 |
| Contrast | 2 | 1 | 0 | 1 |
| Supplement | 30 | 30 | 0 | 0 |
| Cause-Result | 13 | 4 | 7 | 2 |
| Result-Cause | 10 | 8 | 0 | 2 |
| Behavior-Purpose | 4 | 3 | 0 | 1 |
| Purpose-Behavior | 1 | 1 | 0 | 0 |
| Background | 47 | 47 | 0 | 0 |
| Elaboration | 121 | 121 | 0 | 0 |
| Statement-Illustration | 9 | 9 | 0 | 0 |
| Illustration-Statement | 3 | 0 | 3 | 0 |
| Summary | 5 | 2 | 3 | 0 |
| Evaluation | 23 | 15 | 5 | 3 |
| Total | 438 | 242 | 20 | 176 |

Table 6: Corpus statistics of discourse relations and primary-secondary relations. P-S: Primary-Secondary; S-P: Secondary-Primary; MP: Multi-Primary

structure tree construction. In this section, we evaluated our Macro Chinese Discourse Treebank with the tasks of recognition of discourse relation and primary-secondary relation.

We solve the imbalance data problem of primary-secondary relation recognition by re-sampling strategy, that is, re-sampling the data set of nucleus in latter to ten times. A 10 fold cross validation method is adopted, in which sample set is divided into 10 parts of equal size, 9 of them as training set, the remaining one as test set, and different part is selected as test set each time, repeat the exercise 10 times, the average of the 10 results is used to estimate the model performance.

All our classifiers are trained using nltk package with default parameters, and experimented with maximum entropy classifier. Table 7 summarizes the features used in our preliminary experiments.

| Text structural features | | |
|---|---|---|
| Number of paragraphs in Arg1 (Arg2) | | |
| **Text content features** | | |
| Word pairs (top-10 TF-IDF words in Arg1 and Arg2) | | |
| Words and POSs (first three words in Arg1 and Arg2) | | |
| **Semantic features** | | |
| Semantic similarity of Arg1 (Arg2) and title | | |
| **Relation structure features** | | |
| Hierarchical characteristics | | |
| Beginning and end position of Arg1 (Arg2) | | |

Table 7: Features used in our experiments

Table 8 and Table 9 presents the performances of primary-secondary relation recognition and discourse relation recognition. On our corpus, primary-secondary relation recognition achieves F-score of 79.3%, 88.9%, 50.0% absolutely in the three types of nucleus in former, nucleus in latter and multi-nucleus, and discourse relation recognition achieves F-score of 57.1%, 66.7%, 50.0%, 28.6%, 40.0% on the main types of discourse relation. In general, the errors are produced by two causes: 1) imbalanced distribution of the relations; 2) semantic similarity between the relations. The experimental results show that our corpus is helpful to the discourse structure analysis tasks.

| Primary-secondary relation | Precision | Recall | F-score |
|---|---|---|---|
| Primary-Secondary | 72.4% | 87.5% | 79.3% |
| Secondary-Primary | 80.0% | 100.0% | 88.9% |
| Multi-Primary | 85.7% | 35.3% | 50.0% |

Table 8: Results of primary-secondary relation recognition in MCDTB

| Discourse relation | Precision | Recall | F-score |
|---|---|---|---|
| Joint | 61.5% | 53.3% | 57.1% |
| Elaboration | 59.1% | 76.5% | 66.7% |
| Sequence | 100.0% | 33.3% | 50.0% |
| Background | 25.0% | 33.3% | 28.6% |
| Cause-Result | 50.0% | 33.3% | 40.0% |

Table 9: Results of discourse relation recognition in MCDTB

# 6 Conclusion

In this paper, we focus on the problem of discourse primary-secondary relationship whose importance is totally ignored in recent studies. We explore to recognize the primary-secondary relationship which plays a critical role in discourse structure analysis, and view it as an independent task from the discourse rhetorical structure analysis.

In particular, we expand the discourse structure analysis from intra-paragraph to the overall discourse, and propose a macro-micro unified discourse structure representation scheme, and describe the scheme in detail. We also annotate 97 news wire articles based on the representation schema we defined. Evaluation of the corpus resource on discourse relation recognition and primary-secondary relation recognition justifies the effectiveness of the representation schema and corpus resource. In the future work, we will enlarge the scale of the corpus and explore the various discourse patterns.

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
