# Peer review of "Exploring Macro Discourse Structure with Macro-micro Unified Primary-secondary Relationship"

_ACL 2017 — decision unknown_

[Official Review · Reviewer 1 · rating 2 · confidence 4]
soundness 3 · originality 3 · clarity 1 · impact 3 · substance 2 · appropriateness 5 · meaningful comparison 3 · presentation format Poster

This paper proposed a macro discourse structure scheme. The authors carried out
a pilot study annotating a corpus consisting of 97 news articles from Chinese
treebank 8.0. They then built a model to recognize the primary-secondary
relations and 5 discourse relations (joint, elaboration, sequence, background,
cause-result) in this corpus.

The paper is poorly written and I have difficulties to follow it. I strongly
suggest that the authors should find a native English speaker to carefully
proofread the paper. Regarding the content, I have several concerns: 

1 The logic of the paper is not clear and justifiable: 
1) what are "logical semantics" and "pragmatic function"(line 115-116)? I'd
prefer the authors to define them properly.

2) macro discourse structure: there are some conflicts of the definition
between macro structure and micro structure. Figure 4 demonstrates the
combination of macro discourse structure and micro discourse structure. There,
the micro discourse structure is presented *within paragraphs*. However, in the
specific example of micro discourse structure shown in Figure 6, the
micro-level discourse structure is *beyond the paragraph boundary* and captures
the discourse relations across paragraphs. This kind of micro-level discourse
structure is indeed similar to the macro structure proposed by the authors in
Figure 5, and it's also genre independent. So, why can't we just use the
structure in Figure 6? What's the advantage of macro discourse structure
proposed in Figure 5? For me, it's genre dependent and doesn't provide richer
information compared to Figure 6.

By the way, why sentence 6 and sentence 15 are missing in Figure 5? Is it
because they are subtitles? But sentence 12 which is a subtitle is present
there.

2 Corpus construction (section 4) is not informative enough: without a
detailed example, it's hard to know the meaning of "discourse topic, lead,
abstract, paragraph topics (line 627-629)". And you were saying you "explore
the relationships between micro-structure and macro-structure", but I can't
find the correspondent part.

Table 4 is about agreement study The authors claimed "Its very difficult to
achieve high consistence because the judgments of relation and structure are
very subjective. Our measurement data is only taken on the layer of leaf
nodes."--------> First, what are the leaf nodes? In the macro-level, they are
paragraphs; in the micro-level, they are EDUs. Should we report the agreement
study for macro-level and micro-level separately? Second, it seems for me that
the authors only take a subset of data to measure the agreement. This doesn't
reflect the overall quality of the whole corpus, i.e., high agreement on the
leaf nodes annotation doesn't ensure that we will get high agreement on the
non-leaf nodes annotation.

Some other unclear parts in section 4:

Table 4: "discourse structure, discourse relation" are not clear, what is
discourse structure and what is discourse relation? 
Table 5: "amount of macro discourse relations", still not clear to me, you mean
the discourse relations between paragraphs? But in Figure 6, these relations
can exist both between sentences and between paragraphs.

3 Experiments: since the main purpose of the paper is to provide richer
discourse structure (both on macro and micro level), I would expect to see some
initial results in this direction. The current experiment is not very
convincing: a) no strong baselines; b) features are not clearly described and
motivated; c) I don't understand why only a sub set of discourse relations from
Table 6 is chosen to perform the experiment of discourse relation recognition.

In general, I think the paper needs major improvement and currently it is not
ready for acceptance.

[Official Review · Reviewer 2 · rating 1 · confidence 4]
soundness 3 · originality 3 · clarity 3 · impact 3 · substance 1 · appropriateness 5 · meaningful comparison 3 · presentation format Poster

This paper presents a unified annotation that combines macrostructures and RST
structure in Chinese news articles. Essentially, RST structure is adopted for
each paragraph and macrostructure is adopted on top of the paragraphs. 
While the view that nuclearity should not depend on the relation label itself
but also on the context is appealing, I find the paper having major issues in
the annotation and the experiments, detailed below:

- The notion of “primary-secondary” relationship is advocated much in the
paper, but later in the paper that it became clear this is essentially the
notion of nuclearity, extended to macrostructure and making it
context-dependent instead of relation-dependent. Even then, the status
nuclear-nuclear, nuclear-satellite, satellite-nuclear are “redefined” as
new concepts.

- Descriptions of established theories in discourse are often incorrect. For
example, there is rich existing work on pragmatic functions of text but it is
claimed to be something little studied. There are errors in the related work
section, e.g., treating RST and the Chinese Dependency Discourse Treebank as
different as coherence and cohesion; the computational approach subsection
lacking any reference to work after 2013; the performance table of nuclearity
classification confusing prior work for sentence-level and document-level
parsing.

- For the annotation, I find the macro structure annotation description
confusing; furthermore, statistics for the macro labels are not
listed/reported. The agreement calculation is also problematic; the paper
stated that "Our measurement data is only taken on the layer of leaf nodes". I
don't think this can verify the validity of the annotation. There are multiple
mentions in the annotation procedure that says “prelim experiments show this
is a good approach”, but how? Finally it is unclear how the kappa values are
calculated since this is a structured task; is this the same calculation as RST
discourse treebank?

- It is said in the paper that nuclearity status closely associates with the
relation label itself. So what is the baseline performance that just uses the
relation label? Note that some features are not explained at all (e.g., what
are “hierarchical characteristics”?)

- The main contribution of the paper is the combination of macro and micro
structure. However, in the experiments only relations at the micro level are
evaluated; even so, only among 5 handpicked ones. I don't see how this
evaluation can be used to verify the macro side hence supporting the paper.

- The paper contains numerous grammatical errors. Also, there is no text
displayed in Figure 7 to illustrate the example.

[Official Review · Reviewer 3 · rating 3 · confidence 3]
soundness 3 · originality 3 · clarity 4 · impact 3 · substance 4 · appropriateness 5 · meaningful comparison 3 · presentation format Poster

- Strengths:
The macro discourse structure is a useful complement to micro structures like
RST. The release of the dataset would be helpful to a range of NLP
applications.

- Weaknesses:
1. Providing more comparisons with the existed CDTB will be better.
2. The “primary-secondary” relationship is mentioned a lot in this paper,
however, its difference with the nuclearity is unclear and not precisely
defined.
3. The experiment method is not clearly described in the paper.

- General Discussion: